# Relative Handgrip Strength as Marker of Cardiometabolic Risk in Women with Systemic Lupus Erythematosus

**DOI:** 10.3390/ijerph18094630

**Published:** 2021-04-27

**Authors:** Sergio Sola-Rodríguez, José Antonio Vargas-Hitos, Blanca Gavilán-Carrera, Antonio Rosales-Castillo, José Mario Sabio, Alba Hernández-Martínez, Elena Martínez-Rosales, Norberto Ortego-Centeno, Alberto Soriano-Maldonado

**Affiliations:** 1Department of Education, Faculty of Education Sciences, University of Almería, 04120 Almería, Spain; ahm137@ual.es (A.H.-M.); emr809@ual.es (E.M.-R.); asoriano@ual.es (A.S.-M.); 2SPORT Research Group (CTS-1024), CERNEP Research Center, University of Almería, 04120 Almería, Spain; 3Systemic Autoimmune Diseases Unit, Department of Internal Medicine, Virgen de las Nieves University Hospital, 18014 Granada, Spain; joseantoniovh@hotmail.com (J.A.V.-H.); anrocas90@hotmail.com (A.R.-C.); jomasabio@gmail.com (J.M.S.); 4Department of Physical Education and Sport, Faculty of Sport Sciences, University of Granada, 18071 Granada, Spain; bgavilan@ugr.es; 5Systemic Autoimmune Diseases Unit, Department of Internal Medicine, “San Cecilio” University Hospital, 18016 Granada, Spain; nortego@ugr.es

**Keywords:** autoimmune disease, cardiovascular risk, muscle strength, body mass index, metabolism, cardiovascular disease, lupus, risk factors

## Abstract

This study aimed to examine the association of relative handgrip strength (rHGS) with cardiometabolic disease risk factors in women with systemic lupus erythematosus (SLE). Methods: Seventy-seven women with SLE (mean age 43.2, SD 13.8) and clinical stability during the previous six months were included. Handgrip strength was assessed with a digital dynamometer and rHGS was defined as absolute handgrip strength (aHGS) divided by body mass index (BMI). We measured blood pressure, markers of lipid and glucose metabolism, inflammation (high sensitivity C-reactive protein [hs-CRP]), arterial stiffness (pulse wave velocity [PWV]), and renal function. A clustered cardiometabolic risk index (z-score) was computed. Results: Pearson′s bivariate correlations revealed that higher rHGS was associated with lower systolic blood pressure (SBP), triglycerides, hs-CRP, PWV, and lower clustered cardiometabolic risk (r_range_ = from −0.43 to −0.23; all *p* < 0.05). Multivariable linear regression analyses adjusted for age, disease activity (SLEDAI), and accrual damage (SDI) confirmed these results (all *p* < 0.05) except for triglycerides. Conclusions: The findings suggest that higher rHGS is significantly associated with lower cardiometabolic risk in women with SLE.

## 1. Introduction

Systemic lupus erythematosus (SLE) is a chronic autoimmune disease marked with a wide variety of organ system dysfunctions, such as damage to joints, lungs, heart, kidneys, brain, blood vessels or skin [1,2]. The SLE prevalence rates are 20 of every 100,000 women [3], and it affects women at a rate of 10:1 more than men [4]. Due to improved diagnostic methods and treatments [5], mortality in SLE patients continues to improve. However, cardiovascular and metabolic diseases are still one of the biggest causes of mortality in SLE [6], and common risk factors cannot fully explain the increased cardiometabolic risk in this population [7].

Traditional cardiometabolic risk factors including hypertension, diabetes, dyslipidemia, and smoking [8,9], and non-traditional cardiometabolic risk factors including abdominal obesity, insulin resistance, lipid profile, arterial stiffness, renal markers, and high-sensitivity C-reactive protein (hs-CRP; as a marker of inflammation [10,11]) levels [8,12,13] are both expensive and difficult to measure outside a clinical environment [14]. Furthermore, patients with SLE are usually treated with corticosteroids, which at high doses interfere with lipid and glycemic metabolism [15].

Muscular strength is reduced in women with SLE [16,17], and low strength levels are associated with higher fatigue, worse quality of life [18], and higher risk of cardiovascular disease and mortality [19,20]. Handgrip strength, a simple and quick method to assess upper body muscular strength, is inversely associated with coronary heart disease [19,21], inflammation (which appears very often in SLE) [22], and mortality risk [23] in the general population. In women with SLE, handgrip strength is negatively related to obesity [13,17,19], and positively associated with quality of life [24].

Relative handgrip strength (rHGS), defined by the summation of both hands’ strength divided by body mass index (BMI), is an easy instrument for measuring relative muscular strength in clinical practice and public health [25] and has been recommended in recent research to address the increased strength due to body mass [25,26,27,28]. Handgrip strength and BMI have both been linked to cardiometabolic disease risk in the general population [29,30,31,32], although the evidence regarding the association of rHGS with cardiometabolic risk in women is scarce [26]. Since rHGS is cost- and time-efficient, it is of clinical interest to understand the extent to which it might be associated with cardiometabolic risk factors in a population at high risk of cardiometabolic diseases, such as women with SLE.

The primary purpose of the current study was to examine the association of rHGS with biomarkers of cardiometabolic disease risk in women with SLE.

## 2. Materials and Methods

### 2.1. Design and Participants

In this cross-sectional study, a total of 172 Caucasian patients with SLE were invited to participate. Inclusion criteria were: (i) women aged between 18 and 60 years with (ii) >4 SLE classification criteria provided by the American College of Rheumatology [33]; (iii) a minimum follow-up of one year at our unit; and (iv) clinical stability (i.e., the absence of changes in the systemic lupus erythematosus disease activity index (SLEDAI) and/or treatment) during the previous 6 months. Exclusion criteria were: (i) not being able to read, understand, and/or sign the informed consent; (ii) having cancer; (iii) history of clinical cardiovascular disease and/or lung disease in the last year; and (iv) receiving doses of biological treatment higher than 10 mg/d of prednisone (or equivalent) in the previous 6 months. All participants received detailed information about the study aims and procedures and signed informed consent before being included in the study.

### 2.2. Measurement of Relative Handgrip Strength

Muscular strength was assessed through the handgrip strength test. The handgrip strength test [34] was assessed using a digital dynamometer (Model T.K.K.540^®^; Takei Scientific Instruments Co., Ltd., Niigata, Japan) with a precision to the nearest 0.1 kg. Participants performed the trial in a standing position, with the elbow fully extended and the arm relaxed in a neutral position and were encouraged by the evaluators to exert to their maximal effort during a couple of seconds, alternating between the two hands. Participants performed the test twice with a one-minute break between the two attempts of each hand. The aHGS was summed from the best score of each hand. The rHGS was defined as aHGS divided by BMI [25]. Height (cm) was measured using a stadiometer (SECA 222, Hamburg, Germany) and weight (kg) with a bioimpedance device (InBody R20, Biospace, Seoul, Korea). BMI was calculated as weight (kg) divided by height squared (m^2^).

### 2.3. Measurement of Cardiometabolic Risk Factors

Systolic blood pressure (SBP), diastolic blood pressure (DBP), and resting heart rate were measured using the Mobil-O-Graph^®^ 24 h pulse wave analysis monitor (IEM GmbH, Stolberg, Germany) in a sitting position according to the European Society of Hypertension [35], after 5 min of rest.

Arterial stiffness was indirectly assessed through the pulse wave velocity (PWV) [36]. The test was performed in a sitting position after 5 min of rest, using the Mobil-O-Graph^®^ 24 h pulse wave analysis monitor, the operation of which is based on oscillometry recorded by a blood pressure cuff placed on the brachial artery. This instrument is validated for clinical practice [36]. PWV was obtained from a single measurement. The coefficient of variation (CV) of the Mobil-O-Graph for consecutive PWV analyses is 3.4%, and its intraclass correlation coefficient is 0.98 (0.96–0.99) [37].

Venous fasting blood samples were collected in the morning with heparin as the anticoagulant. Blood was centrifuged at 3500 rpm for 15 min to separate the plasma, which was subsequently removed. Plasma triglycerides, high-density lipoprotein cholesterol (HDL-c), low-density lipoprotein cholesterol (LDL-c), total cholesterol, glucose, urea, albumin and creatinine concentrations were analyzed enzymatically with an autoanalyzer (Olympus Diagnostic, Hamburg, Germany). Insulin was measured with an enzyme immunoassay kit, and the homeostasis model assessment of insulin resistance (HOMA-IR) was calculated [(fasting insulin (μIU/mL) × fasting glucose (mg/dL))/405]. Apolipoproteins A and B, hs-CRP, and glycosylated hemoglobin were determined by immunoturbidimetry (HORIBA-ABX Diagnostics, Japan) with an autoanalyzer (PENTRA-400, HORIBA-ABX Diagnostics, Japan). The albumin-creatinine ratio was measured from a first-morning urine sample. Values above or equal to 30 mg/g in women were considered pathological. The estimated glomerular filtration rate was determined by the modification of diet in renal disease (MDRD) equation [38]: (GFe (MDRD)):175 × SCr − 1.154 × age − 0.203 × 0.742 

SCr: serum creatinine

### 2.4. Other Measurements

All participants filled out a sociodemographic and clinical data questionnaire to gather information, such as age, disease duration, current medication (including antidiabetics and corticosteroids), and tobacco consumption. The systemic lupus erythematosus disease activity index (SLEDAI) was included to assess disease activity [39], considering the presence or absence of several clinical and analytical manifestations in the preceding 10 days. The final score ranges from 0 to 105, where a higher score indicates a higher degree of disease activity. The degree of tissue damage from the onset of the disease was evaluated by the International Collaborating Clinics/American College of Rheumatology’s systemic lupus damage index (SLICC-SDI) [40]. The score ranges from 0 to 40, where a higher score indicates greater damage produced by SLE in the last 6 months.

### 2.5. Sample Size

The sample size was calculated for a clinical trial evaluating the effects of aerobic exercise on arterial stiffness, inflammation, and fitness, which was published earlier [41]. We recruited 58 participants for that trial, although a larger sample (*n =* 77) was used to perform baseline evaluations for cross-sectional analyses.

### 2.6. Statistical Analysis

The descriptive characteristics of the study participants are presented as means and standard deviations for continuous variables, and as frequencies and percentages for categorical variables, unless otherwise indicated in Table 1. Due to the presence of outliers, hs-CRP was winsorized. Normality was assessed through histograms, the Kolmogorov–Smirnov Test, and Q–Q plots, with muscular strength and cardiometabolic risk factors showing a normal distribution. Pearson’s bivariate correlations were used to explore the raw association between rHGS and cardiometabolic risk factors, and we additionally assessed the crude association of aHGS and BMI with cardiometabolic risk factors. Regression models were built including each cardiometabolic risk factor as dependent variables in separate models. rHGS, age, SLEDAI, and SDI were entered as independent variables in all models (enter method). Age, SLEDAI, and SDI were entered as covariables due to their potential role as confounders [42]. Menopause, statins or corticosteroids were initially included, but they did not alter the coefficients, and thus they were not included in final models to avoid overfitting [43].

A clustered cardiometabolic risk index (z-score) [12] was created using the mean of the standardized scores [(value-mean)/standard deviation] for SBP, fasting glucose, triglycerides, HOMA-IR, total cholesterol/HDL-c, and hs-CRP. Statistical significance was set at *p* < 0.05.

## 3. Results

The flowchart of the study participants is presented in Figure 1. From a total of 172 patients initially invited, 81 refused to participate (41 patients reported living very far from the hospital, 36 were not able to find time to perform the evaluations, and 4 were not interested), 12 patients did not present clinical stability during the previous 6 months to the beginning of the study, and 2 patients had cardiovascular disease during the previous year. A total of 77 women with SLE (mean age 43.2, SD 13.8) complied with the inclusion criteria, agreed to participate, and were assessed in two waves (49 women in October 2016 and 28 women in February 2017). Both evaluations were identical. Two women did not perform the handgrip strength test due to a wrist injury.

The descriptive characteristics of the study participants are presented in Table 1. The average BMI was 25.5 (SD 0.51) kg/m^2^. The average aHGS was 47.2 (SD 1.24) kg and for rHGS was 1.89 (SD 0.05) units. Regarding cardiometabolic risk variables, the average SBP was 118 (SD 1.29) mmHg, the average DBP was 76.5 (SD 1.18) mmHg, and the average fasting glucose levels were 76.3 (SD 2.17) mg/dL. Average total cholesterol was 177.5 (SD 3.56) mg/dL, the average hs-CRP levels were 2.73 (SD 0.35) mg/L and the average PWV was 6.47 (SD 0.17) m/s.

Table 2 represents the raw association of rHGS, aHGS, and BMI with cardiometabolic risk factors. rHGS was negatively associated with SBP, triglycerides, hs-CRP, PWV, and z-score (r_range_ = from –0.43 to −0.23; all *p* < 0.05). aHGS was negatively associated with triglycerides and PWV (r_range_ = from −0.34 to −0.23; all *p* < 0.05). Finally, BMI was positively associated with SBP, DBP, fasting glucose, HOMA-IR, PWV, and z-score (r_range_ = from 0.23 to 0.44; all *p* < 0.05). A graphic representation of the crude association of rHGS and cardiometabolic risk factors is presented in Figure 2. The linear regression models evaluating the association of rHGS and cardiometabolic risk factors are presented in Table 3. rHGS was inversely associated with SBP (unstandardized coefficient (B) = −6.58; 95% confidence interval (CI) −11.91 to −1.26; *p* = 0.016), hs-CRP (B = −1.67; 95% CI −3.11 to −0.23; *p* = 0.023), PWV (B = −0.34; 95% CI −0.58 to −0.09; *p* = 0.007) and z-score (B = −0.30; 95% CI −0.54 to −0.06; *p* = 0.014). These results were consistent even when statins and corticosteroids were included as covariates.

## 4. Discussion

The main finding of this study is that a higher rHGS was associated with lower SBP, triglycerides, hs-CRP, PWV, and clustered cardiometabolic risk index (z-score) in women with SLE. Furthermore, rHGS could be an alternative to aHGS when evaluating cardiometabolic risk. Our results were consistent despite adjusting for multiple potential confounders such as age, SLEDAI, SDI, statins, menopause, smoking or corticosteroids.

The association of aHGS and cardiometabolic risk has been previously studied in the general population. Lee et al. [27] found that a higher aHGS was associated with lower cardiovascular risk in older Korean adults. Similar findings were described by Leong et al. [21], who found that aHGS was inversely associated with all-cause death in a prospective cohort study with 140,000 men and women. However, Gregorio-Arenas et al. [44] found no association of aHGS with cardiometabolic risk in a sample of 228 perimenopausal women. In line with this, Gubelmann, Vollenweider and Marques-Vidal [45] observed no association between aHGS and cardiovascular risk in healthy adults. Regarding rHGS, previous studies have assessed its association with cardiometabolic risk, although not in rheumatological or autoimmune populations. Choquette et al. found that rHGS could be an indicator of cardiometabolic risk in 1793 community-dwelling men and women [25]. Moreover, Lawman et al. [28] found that higher rHGS was significantly associated with lower SBP, triglycerides, glucose, and higher HDL in both healthy men and women. Finally, Campa et al. [46] demonstrated that resistance training is effective in improving both cardiometabolic risk factors and rHGS in obese women, but improvements regarding rHGS are only achieved if training frequency is high and prolonged over time [47]. Our results are overall in line with these findings derived from other populations and extend current knowledge on potential indicators of cardiometabolic risk in SLE, as well as agreeing with recent literature.

The novel approach of this study is the concurrent analysis of the association of rHGS, aHGS and BMI itself with cardiometabolic risk factors. Although no statistical test can compare the strength of their independent association with the outcomes, these analyses provide the opportunity to determine which of these markers of risk is more worthwhile in clinical practice. Overall, rHGS and BMI were clearly better indicators of cardiometabolic risk than aHGS. However, when comparing BMI with rHGS, the results were less clear. While BMI was associated with markers of insulin resistance and the association with the clustered cardiometabolic risk score was stronger than with rHGS, rHGS was more strongly associated with arterial stiffness and, more importantly, with hs-CRP. As inflammation is a hallmark of autoimmune diseases including SLE, these results should not be taken into consideration when deciding whether to include the assessment of handgrip strength in clinical practice. The relatively low sample size precludes making strong arguments either in favor of or against this, although further research on this topic seems warranted. In practical terms, it is obvious that BMI is the simplest way to obtain a strong marker of cardiometabolic risk. However, it must be considered that adding a handgrip strength assessment takes approximately 2 min (including double assessment of both hands), which, depending on the context, might be feasible or not.

This study has potential limitations. Although other widely used tools to measure CV risk have been proposed, these tools could underestimate CV risk in patients with SLE. Our study provides a greater knowledge of CV risk using individual factors and a cluster score. The cross-sectional design precludes the establishment of causal relationships; therefore, our results must be corroborated in future prospective and experimental research. The sample size was relatively small, and we do not know whether these results apply to men or to women with medium or high disease activity, as only women with mild disease activity were included.

## 5. Conclusions

The findings suggest that higher rHGS is significantly associated with lower cardiometabolic risk in women with SLE. Although assessing rHGS might add relevant information regarding the potential cardiometabolic risk of SLE patients, BMI alone is a rather good indicator of cardiometabolic risk that might be preferred under time-constrained situations.

## Figures and Tables

**Figure 1 ijerph-18-04630-f001:**
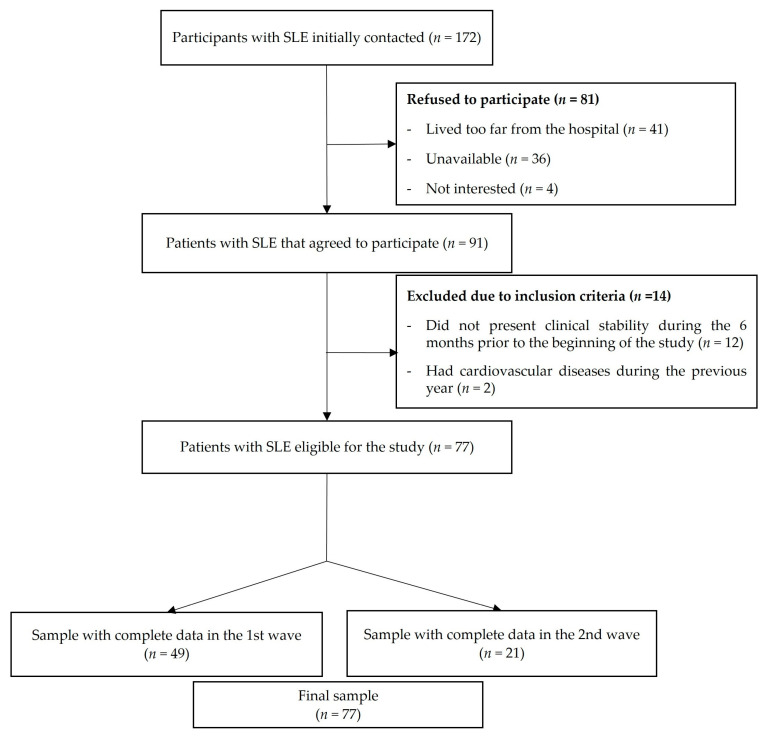
Flow diagram of the inclusion of women with systemic lupus erythematosus (SLE) for the present study.

**Figure 2 ijerph-18-04630-f002:**
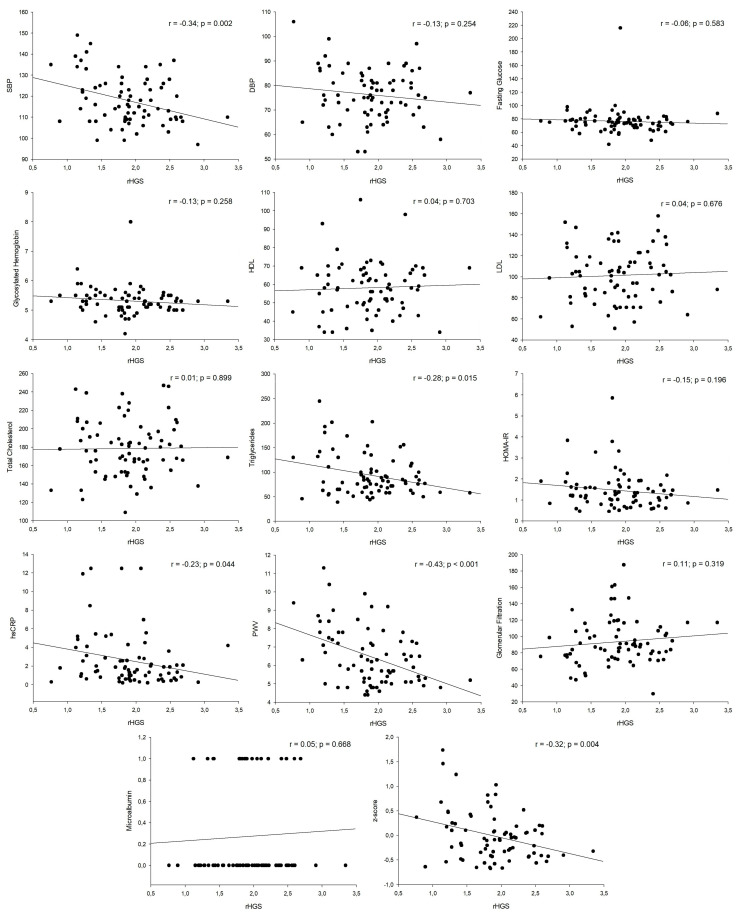
Graphic representation of the crude association of rHGS and cardiometabolic risk factors.

**Table 1 ijerph-18-04630-t001:** Descriptive characteristics of the study participants (*n =* 77).

	Mean	SD
Age (years)	43.2	1.57
Weight (kg)	65.1	1.27
Height (cm)	160.1	0.77
Body Mass Index (kg/m^2^)	25.5	0.51
Absolute Handgrip Strength (kg)	47.2	1.24
Relative Handgrip Strength (kg/BMI)	1.89	0.05
SLEDAI	0.6	0.17
Duration of SLE (years)	13.9	1.15
Systolic Blood Pressure (mmHg)	118	1.29
Diastolic Blood Pressure (mmHg)	76.5	1.18
Pulse Wave Velocity (m/s)	6.47	0.17
Fasting Glucose (mg/dL)	76.3	2.17
Glycosylated Hemoglobin (%)	5.31	
High Density Lipoprotein (mg/dL)	57.8	1.57
Low Density Lipoprotein (mg/dL)	100.7	2.88
Total Cholesterol (mg/dL)	177.5	3.56
Triglycerides (mg/dL)	93.6	4.85
Homeostatic Model Assessment	1.45	0.09
hs-CRP (mg/L)	2.73	0.17
Glomerular Filtration (mL/min/1.73 m^2^)	92.6	3.33
Microalbuminuria (%)	28	
Cumulative Prednisone dose (mg)	2875	2677
Daily Prednisone dose (mg)	3.99	0.57
Prednisone use (%)	65	
Immunosuppressants (%)	45	
Antimalarials (%)	89	

For absolute and relative handgrip strength the total sample size was *n =* 75 due to missing data. SLEDAI: systemic lupus erythematosus disease activity index; hs-CRP: high-sensitivity C-reactive protein.

**Table 2 ijerph-18-04630-t002:** Pearson’s bivariate correlations analysis evaluating the raw association between relative handgrip strength, absolute handgrip strength and body mass index with cardiometabolic risk components in women with systemic lupus erythematosus.

	rHGS (*n =* 75)	aHGS (*n =* 75)	BMI
SBP	−0.34 **	−0.15	0.40 **
DBP	−0.13	0.01	0.32 **
Fasting Glucose	−0.06	0.08	0.23 *
Glycosylated Hemoglobin	−0.13	−0.07	0.11
HDL	0.04	0.08	0.04
LDL	0.04	0.04	−0.00
Total Cholesterol	0.01	0.03	0.04
Triglycerides	−0.28 *	−0.23 *	0.15
HOMA-IR	−0.15	0.11	0.43 **
hs-CRP	−0.23 *	−0.15	0.17
PWV	−0.43 **	−0.34 **	0.24 *
Glomerular Filtration	0.11	0.08	−0.10
Microalbumin	0.05	−0.04	−0.15
z−score	−0.32 **	−0.09	0.44 **

SBP: systolic blood pressure; DBP: diastolic blood pressure; HDL: high-density lipoprotein; LDL: low-density lipoprotein; HOMA-IR: homeostatic model assessment of insulin resistance; hs-CRP: high-sensitivity C-reactive protein; PWV: pulse wave velocity. Notes: * *p* < 0.05; ** *p* < 0.01.

**Table 3 ijerph-18-04630-t003:** Multivariable linear regression analysis evaluating the association of relative handgrip strength with cardiometabolic risk components in women with systemic lupus erythematosus (*n =* 75).

	Beta	B	Std Error	95% CI		*p*	R^2^
SBP	−0.29	−6.58	2.67	−11.91	−1.26	**0.016**	0.20
DBP	−0.10	−2.02	2.63	−7.27	3.23	0.445	0.03
Fasting Glucose	−0.09	−3.58	5.00	−13.55	6.39	0.476	0.01
Glycosylated Hemoglobin	−0.02	−0.02	0.11	−0.25	0.20	0.846	0.10
HDL	0.10	2.77	3.56	−4.33	9.89	0.438	0.02
LDL	0.16	8.06	6.08	−4.06	20.20	0.189	0.14
Total Cholesterol	0.15	9.03	7.25	−5.44	23.50	0.218	0.18
Triglycerides	−0.23	−19.41	10.50	−40.35	1.52	0.069	0.12
HOMA-IR	−0.19	−0.34	0.22	−0.79	0.10	0.127	0.03
hs-CRP	−0.29	−1.67	0.72	−3.11	−0.23	**0.023**	0.09
PWV	−0.11	−0.34	0.12	−0.58	−0.09	**0.007**	0.91
Glomerular Filtration	−0.14	−7.68	5.75	−19.16	3.80	0.187	0.37
Microalbumin	−0.11	−0.01	0.11	−0.23	0.21	0.925	0.10
z-score	−0.30	−0.30	0.12	−0.54	−0.06	**0.014**	0.15

B: unstandardized coefficient; SBP: systolic blood pressure DBP: diastolic blood pressure; HDL: high-density lipoprotein; LDL: low-density lipoprotein; HOMA-IR: homeostatic model assessment of insulin resistance; hs-CRP: high-sensitivity C-reactive protein; PWV: pulse wave velocity. All regression models were adjusted for age, SLEDAI, and SDI. Regression models were built including each cardiometabolic risk factor as dependent variables in separate models. Relative handgrip strength was entered as the independent variable in all models (enter method) where age, SLEDAI, and SDI were entered as confounders in order to adjust the independent variable. Statistically significant associations (*p* < 0.05) are highlighted in bold.

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
