# Peer review of "Relative Handgrip Strength as Marker of Cardiometabolic Risk in Women with Systemic Lupus Erythematosus"

_ijerph, 2021, doi:10.3390/ijerph18094630_

Round 1
Reviewer 1 Report
This manuscript investigated the association of relative handgrip strength with cardiometabolic disease risk factors in women with systemic lupus erythematosus. I have some minor/major revisions to suggest in order to improve the manuscript quality:
Abstract: Line 21: Replace 77 with Seventy-seven and 6 with six. What is the point of performing a regression analysis without having considered any confounding variable? just reading the statistical analysis section I realize that Age, SLEDAI, and SDI have been included as co-variables. However, this is not clear in the abstract and a sentence such as: Relative handgrip strength remains a significant predictor of … even adjusting for Age, SLEDAI, and SDI. I guess none of the women were in menopause, otherwise, this variable should also be included.
Keywords: Remove words already included in the title
Introduction:
The rationality of the study was well explained. Congrats
Methods
How was the sample size calculated? The result of the power analysis should be reported.
Table 2. No values should be highlighted
Table 3. It should be specified that independent variables have been adjusted for confounding parameters. Also, R2 for each model should be included.
The manuscript largely lacks illustrative representation by summarizing correlations in tables without showing the original data but rather an r/r2/p value. This is a poor representation of scientific data.
The discussion section is very descriptive and offers limited comparisons to previous research. In this regard, the MDPI literature should be consulted as recently innovative studies have been proposed regarding the variables considered. Also, physical activity as a solution for improving HS should be considered. In one of the two studies reported below, the associations between handgrip strength and risk factors have already been analyzed. How does this study differ? What is the additional value? This part should be stressed Here are some useful titles for improving this section:
- Int J Environ Res Public Health. 2020;17(4): 1163. Comparison of the Effect of Different Resistance Training Frequencies on Phase Angle and Handgrip Strength in Obese Women: A Randomized Controlled Trial doi: 10.3390/ijerph17041163
- J. Funct. Morphol. Kinesiol. 2020, 5(3), 51; Effects of Different Resistance Training Frequencies on Body Composition, Cardiometabolic Risk Factors, and Handgrip Strength in Overweight and Obese Women: A Randomized Controlled Trial. https://doi.org/10.3390/jfmk5030051
Author Response
Reviewer #1
This manuscript investigated the association of relative handgrip strength with cardiometabolic disease risk factors in women with systemic lupus erythematosus. I have some minor/major revisions to suggest in order to improve the manuscript quality:
Abstract: Line 21: Replace 77 with Seventy-seven and 6 with six. What is the point of performing a regression analysis without having considered any confounding variable? just reading the statistical analysis section I realize that Age, SLEDAI, and SDI have been included as co-variables. However, this is not clear in the abstract and a sentence such as: Relative handgrip strength remains a significant predictor of … even adjusting for Age, SLEDAI, and SDI. I guess none of the women were in menopause, otherwise, this variable should also be included.
Response: Information regarding co-variables has been included in abstract. There were a total of 30 (39%) women in menopause. This variable was considered a potential confounder that was included into the models. As it had no influence in the regression coefficients, menopause was not finally included to avoid overfitting. This has been indicated in 2.6 Statistical Analysis.
Reviewer comment: Keywords: Remove words already included in the title
Response: Done. The keywords have been changed by other relevant terms.
Reviewer comment: Introduction: The rationality of the study was well explained. Congrats
Response: Thank you very much for your positive feedback.
Reviewer comment: Methods: How was the sample size calculated? The result of the power analysis should be reported.
Response: The sample size was calculated for a clinical trial about the effects of aerobic exercise on arterial stiffness, inflammation and fitness that was published earlier [https://www.ncbi.nlm.nih.gov/pmc/articles/PMC6306776/]. We recruited 58 participants for that trial, although a larger sample (n=77) was recruited for for cross-sectional analyses. This information has been added in paragraph 2.5.
Reviewer comment: Table 2. No values should be highlighted.
Response: Done.
Reviewer comment: Table 3. It should be specified that independent variables have been adjusted for confounding parameters. Also, R2 for each model should be included.
Response: That information requested by the reviewer in Table 3 has been included as footnotes, and the R2 for each model is now presented.
Reviewer comment: The manuscript largely lacks illustrative representation by summarizing correlations in tables without showing the original data but rather an r/r2/p value. This is a poor representation of scientific data.
Response: Following the reviewer’s suggestion, we have included a graphical representation of the results (Figure S1). We believe that enhances the presentation of the results in a more illustrative way.
Reviewer comment: The discussion section is very descriptive and offers limited comparisons to previous research. In this regard, the MDPI literature should be consulted as recently innovative studies have been proposed regarding the variables considered. Also, physical activity as a solution for improving HS should be considered. In one of the two studies reported below, the associations between handgrip strength and risk factors have already been analyzed. How does this study differ? What is the additional value? This part should be stressed Here are some useful titles for improving this section:
- Int J Environ Res Public Health. 2020;17(4): 1163. Comparison of the Effect of Different Resistance Training Frequencies on Phase Angle and Handgrip Strength in Obese Women: A Randomized Controlled Trial doi: 10.3390/ijerph17041163
- J. Funct. Morphol. Kinesiol. 2020, 5(3), 51; Effects of Different Resistance Training Frequencies on Body Composition, Cardiometabolic Risk Factors, and Handgrip Strength in Overweight and Obese Women: A Randomized Controlled Trial. https://doi.org/10.3390/jfmk5030051
Response: Thank you very much for providing us with those titles. Both studies have been cited and discussed in paragraph 4. Discussion. In addition, the discussion has been revised to account for the reviewer concern and offer further comparisons with previous research.
Reviewer 2 Report
Authors reported a research article with the aim to elucidate the association of relative handgrip strength (rHGS) with cardiometabolic disease risk factors in women with systemic lupus erythematosus. The recruted in the study 77 women with SLE in clinical stability setting. The main issue from the study was the following "The findings suggest that higher rHGS is significantly associated with lower cardiometabolic risk in women with SLE." The weaknesses of the study were low sample sizeand a lack of methodological approach to rule-out low physical endurence and possible negative impact of corticosteroind on muscle mass and activity. The strength of the study was a novelity in metabolic risk determination among SLE women. The aim is clear and coincides with the methods and results. The tables and figures are legible and logically structured. Although the findings are interested, there are several items need to be explained.
- Authors measured cardiometabolic risk factors, while for SLE patients older 40 years they could estemate ASCVD risk. Please, check and explaine.
- Authors would compare cardiometabolic risk model with ASCVD risk model. Please, give clear explanation.
- There are no data about concomitant medication and a duration of previous GCS period to treat. Please, check and explain whether GCS could be a trigger for low rHGS test.
- Please. add issues mentioned above to the section Study limitation and discuss them widely.
Author Response
Reviewer #2
Reviewer comment: Authors reported a research article with the aim to elucidate the association of relative handgrip strength (rHGS) with cardiometabolic disease risk factors in women with systemic lupus erythematosus. The recruted in the study 77 women with SLE in clinical stability setting. The main issue from the study was the following "The findings suggest that higher rHGS is significantly associated with lower cardiometabolic risk in women with SLE." The weaknesses of the study were low sample size and a lack of methodological approach to rule-out low physical endurence and possible negative impact of corticosteroid on muscle mass and activity. The strength of the study was a novelity in metabolic risk determination among SLE women. The aim is clear and coincides with the methods and results. The tables and figures are legible and logically structured. Although the findings are interested, there are several items need to be explained.
Response: Thank you very much for the positive evaluation of our manuscript.
Reviewer comment: Authors measured cardiometabolic risk factors, while for SLE patients older 40 years they could estimate ASCVD risk. Please, check and explain. Authors would compare cardiometabolic risk model with ASCVD risk model. Please, give clear explanation.
Response: Thank you for this relevant comment. As the reviewer suggests, we were initially thinking of including some ASCVD risk score, as we strongly believe this kind of estimations give important information both from a clinical and research prespective. However, we also considered that i) only participants 40 years old or over could be included, thus considerably reducing the sample size, which is already relatively low; ii) these algorithms have shown to underestimate the CVD risk in patients with SLE and it is currently recommended to estimate the individual risk. We recently discussed about these issues in a letter to the editor (https://pubmed.ncbi.nlm.nih.gov/32940697/) regarding a relevant paper in the field. Therefore, we finally decided not to include the ASCVD risk model, although we will definitely consider it for current ongoing research.
Reviewer comment: There are no data about concomitant medication and a duration of previous GCS period to treat.
Response: Information on the types and frequency of treatment for patients (including corticosteroids) has been included in Table 1. Usually, in relation to corticosteroids, the information is usually expressed as a percentage of patients taking it and the mean dose in mg/d of prednisone.
Reviewer comment: Please, check and explain whether GCS could be a trigger for low rHGS test.
Response: We thank the reviewer for raising this relevant point. The use of GCS has been associated with muscle strength decrease but, normally, when taken at high doses and/or for prolonged periods of time (https://pubmed.ncbi.nlm.nih.gov/33086134/). However, given that the level of disease activity was generally low among our participants, the use and doses of GCS was low. Therefore, we consider that the influence of GCS on our results are minimum. In this sense, our results were consistent despite adjusting for GCS.
Reviewer comment: Please add issues mentioned above to the section Study limitation and discuss them widely.
Response: All the issues mentioned above have been carefully addressed. Thank you very much for the time to review our manuscript.